# Familial Hypercholesterolemia and Its Current Diagnostics and Treatment Possibilities: A Literature Analysis

**DOI:** 10.3390/medicina58111665

**Published:** 2022-11-17

**Authors:** Kristina Zubielienė, Gintarė Valterytė, Neda Jonaitienė, Diana Žaliaduonytė, Vytautas Zabiela

**Affiliations:** 1Department of Cardiology, Lithuanian University of Health Sciences Kaunas Clinics, LT-50161 Kaunas, Lithuania; 2Department of Cardiology, Lithuanian University of Health Sciences Kaunas Hospital, LT-45130 Kaunas, Lithuania; 3Kaunas Region Society of Cardiology, LT-44307, Kaunas, Lithuania; 4Institute of Cardiology Kaunas, Cardiology Research Automation Laboratory, Lithuanian University of Health Sciences, LT-44307 Kaunas, Lithuania

**Keywords:** familial hypercholesterolemia, Dutch Lipid Clinic Network diagnostic criteria, risk factors, coronary artery disease

## Abstract

Familial hypercholesterolemia (FH) is a common, inherited disorder of cholesterol metabolism. This pathology is usually an autosomal dominant disorder and is caused by inherited mutations in the APOB, LDLR, and PCSK9 genes. Patients can have a homozygous or a heterozygous genotype, which determines the severity of the disease and the onset age of cardiovascular disease (CVD) manifestations. The incidence of heterozygous FH is 1: 200–250, whereas that of homozygous FH is 1: 100.000–160.000. Unfortunately, FH is often diagnosed too late and after the occurrence of a major coronary event. FH may be suspected in patients with elevated blood low-density lipoprotein cholesterol (LDL-C) levels. Moreover, there are other criteria that help to diagnose FH. For instance, the Dutch Lipid Clinical Criteria are a helpful diagnostic tool that is used to diagnose FH. FH often leads to the development of early cardiovascular disease and increases the risk of sudden cardiac death. Therefore, early diagnosis and treatment of this disease is very important. Statins, ezetimibe, bile acid sequestrants, niacin, PCSK9 inhibitors (evolocumab and alirocumab), small-interfering-RNA-based therapeutics (inclisiran), lomitapide, mipomersen, and LDL apheresis are several of the available treatment possibilities that lower LDL-C levels. It is important to say that the timeous lowering of LDL-C levels can reduce the risk of cardiovascular events and mortality in patients with FH. Therefore, it is essential to increase awareness of FH in order to reduce the burden of acute coronary syndrome (ACS).

## 1. Introduction

One of the most important risk factors for CVD is an increase in cholesterol levels in the blood [1]. An increase in cholesterol levels is a condition that does not show any signs or symptoms; however, it leads to an increase in the risk of atherosclerotic disease and, subsequently, premature death [2]. One of the reasons for the increase in blood cholesterol is FH. FH is a genetic disease that manifests as a disorder of cholesterol metabolism [3]. FH is caused by mutations in hereditary genes (APOB, LDL-R, PCSK9) in an autosomal dominant (AD) manner [4,5]. If FH is caused by a mutation in the adapter protein 1 (LDL-R AP1) gene, the disease is inherited in an autosomal recessive manner [6]. Heterozygous and homozygous forms of FH are allocated by genotype [7]. Elevated low-density lipoprotein cholesterol concentration has been detected in these patients from birth, which in the case of heterozygous FH, ranges from 8 to 15 mmol/L, while homozygous FH is characterized by an increase in LDL-C to 12–30 mmol/L. An increase in LDL-C eventually leads to atherosclerosis and its clinical manifestations [8].

## 2. Epidemiology

FH is relatively common in comparison to other genetic disorders, and affects an estimated 20 million people in the world. However, 90% of those patients are underdiagnosed [9]. Prevalence varies according to geographical location and the diagnostic criteria used. Moreover, there is an absence of centralized registries and no agreement on unified diagnostic criteria globally, which makes the evaluation and comparison of FH epidemiology in different countries even more difficult. However, in a recent meta-analysis, the prevalence of FH in Asia was 0.19% (4 studies), in Europe—0.32% (19 studies), and 0.32% in North America (9 studies) [10]. In Russia, using the Dutch Lipid Clinic Network criteria, the prevalence of patients with definite or probable heterozygous FH combined was 0.58% [11]. Based on systematic reviews and meta-analyses, the overall prevalence of FH ranges from about 1: 200 to 250 in the heterozygous FH group, and from 1: 100.000 to 160.000 in the homozygous FH group [12].

The prevalence of FH increases with the additional ischemic heart disease (IHD) or premature ischemic heart disease (PIHD) diagnosis. In comparison, 3.57% with IHD vs. 5.43% with PIHD in Asia, 2.26% vs. 8.04% in Europe, and 4.38% vs. 2.83% in North America, respectively [13]. Furthermore, a study on severe hypercholesterolemia (SH) has shown that SH is an independent risk factor for incident coronary heart disease (CHD) and cerebrovascular disease [14].

Data collected in Vilnius University Hospital Santaros Clinics revealed that dyslipidemia was detected in 90.1% patients with IHD [14]. After comparison with epidemiological data, the prevalence FH in Lithuania could be up to 1:10,000 [15].

## 3. Etiology

When a patient has FH, hyperlipidemia occurs since childhood, and the risk of developing early-onset IHD is significantly higher compared to polygenic forms of hypercholesterolemia [16].

FH is an autosomal dominant disorder, and can be caused by a variety of gene defects: LDL-R, APOB, and PCSK9 [17]. Very rare mutations in the APOE and STAP1 genes may also be a cause of FH [18], and a mutation in the LDL-R AP1 gene may be a cause of autosomal recessive FH [19].

Studies have shown that FH is most commonly (52–76%) caused by mutations in the LDL-R gene, mutations in the APOB gene in 2 to 10% of cases, and PCSK9—in up to 2% of patients [13]. Up to 20–40% mutations are not identified in patients with a clinically definite or possible diagnosis of FH [20].

## 4. Pathogenesis of FH

The most common cause of FH, accounting for 80–85% of FH cases, is a mutation in the LDL-R gene [21]. In this case, FH is caused by impaired LDL-R synthesis or impaired receptor binding, which leads to an excessive concentration of LDL-C in the blood [22]. When there are excessive levels of low-density lipoprotein (LDL) or modified LDL in the blood, macrophages phagocytose these lipoproteins and turn them into the foamy cells that form atherosclerotic lesions. [23]

The second most common mutation is a mutation in APOB100, causing 5–10% of cases [24]. APOB is a ligand that encodes the one apolipoprotein included in LDL. Causative variants produce a protein unable to bind LDLR [25]. Mutation in the PCSK9 gene is rare (approximately 2%). PCSK9 variants can either be loss-of-function variants, generating less functional proteins, or gain-of-function variants, producing more active proteins [26]. Both mechanisms lead to a reduced expression of LDLR resulting in plasma LDL accumulation.

When mutations in the abovementioned genes are not identified, but the patient has a phenotypical FH expression, in these cases it could be mutation in rare FH determined genes or polygenic inheritance can be suspected.

## 5. Familial Hypercholesterolemia and Risk of Major Cerebro-Cardiovascular Event (MACCE)

Patients with FH are exposed to high cholesterol levels in blood from birth. Therefore, they are more inclined to the development of atherosclerotic lesions in the heart, brain, and peripheral arteries, leading to an increased risk of PIHD or cerebrovascular disease (CVD). In a recent meta-analysis (18 studies) it was found that the risk of cardiovascular events (CVE) and death was significantly increased in patients with FH. The study showed that cardiovascular events can be predicted by clinical or genetic diagnostic criteria as they had a similar ability to do it. The Dutch Lipid Clinic Diagnostic Criteria (DLCN) ≥ 6 points (diagnosis of definite or probable FH) and DLCN < 3 points (diagnosis of unlikely FH) had the highest risk of CVE prediction ability [27]. Moreover, another study has shown that FH was associated with increased risk of MACCE, independently of age, sex, smoking, body mass index, hypertension, or diabetes mellitus (HR = 2.30, 95%CI = 1.09 to 4.84, *p* = 0.028) [28]. Besides FH diagnosis, LDL-cholesterol level can also increase the risk of cardiovascular events. A recent study compared the LDL-cholesterol levels in patients after an ischemic stroke and those patients with higher LDL cholesterol levels in the blood had higher risk of cardiovascular events [29].

## 6. Lipidogram Rates

The Canadian Cardiovascular Society (CCS) dyslipidemia guidelines and FH position statement recommend global screening of cholesterol levels in blood in all men > 40 years and all women > 50 years or after menopause. In children with a first-degree family history of atherosclerotic cardiovascular disease or FH, it is recommended to consider to monitor cholesterol level in the blood at age 11 years or as early as 12 months [30].

According to the 2021 ESC guidelines on CVD prevention in clinical practice, the SCORE2 algorithm cannot be used for patients with a genetic lipid disorder, such as FH. Specific LDL-C thresholds and targets are recommended irrespective of estimated CV risk for patients with FH or other rare/genetic lipid disorders.

## 7. Screening for FH

FH is a disease that can be screened according to the standards of the World Health Organization (WHO) screening programs, but in many countries, the diagnosis of this disease is inadequate. The disease is most commonly identified and diagnosed based on history (early cases of IHD or cases of FH in the family) and an increase in total cholesterol (TC) and LDL-C at a young age in blood [31]. Once a case of FH has been confirmed in the family, a step-by-step examination of family members and identification of all patients is required [32]. Cascade screening (CS), whereby family members are traced from an established FH index case, is more cost effective than any other screening strategy currently available [6], and is recommended in NICE guidelines. Approximately half of the first-degree relatives of an index case will be found to have the FH mutation [6].

## 8. Diagnosis of FH

FH can be inspected using different diagnostic systems. The criteria of the Dutch Lipid Clinic (Table 1) are the most widely used in clinical practice [33]. Based on these criteria, FH is diagnosed referring to blood lipid levels, family history of IHD (early death from myocardial infarction (MI) or other early IHD), patient’s history of early coronary artery disease (CAD), objective examination (tendon xanthoma, arcus cornealis), a and naive lipidogram.

The most cost-effective way to identify FH is a stepwise screening of proband family members [34]. Patients should be referred to lipid clinics (or tertiary level hospitals) if their TC concentration is > 7.5 mmol/L (necessarily if > 8 mmol/L) or LDL > 5.5 mmol/L (required > 6 mmol/L) for detailed examination and genetic testing [35].

A total of 2104 unique variants were identified as being associated with FH, but only 166 variants have been proven by complete in vitro functional studies to be causative of disease [36]. FH can be caused by inherited mutations in the LDLR, APOB, and PCSK9 genes. About 60–80% of people with FH have a mutation found in one of these three genes, but in some probands, the FH phenotype is associated with variants of other genes [37]. The typical clinical picture of FH can result from the accumulation of common cholesterol-increasing alleles (polygenic FH) [37].

According to a publicly available analysis, it is found that LDLR, APOB and PCSK9 variants are associated with familial hypercholesterolemia. The largest number of identified variants were found in Europe (*n* = 1491), followed by Asia (*n* = 332), America (*n* = 265), Oceania (*n* = 134), and Africa (*n* = 77). No *APOB* or PCSK9 variants were identified in African countries [36].

Gene mutations are discovered in only 60–70% of individuals diagnosed with definite or probable FH according to clinical criteria [38]. This means that some cases of FH are of polygenic origin or cause mutations in genes that have not been identified yet [39]. The possibility of secondary hyperlipidemia (usually differentiated with hypothyroidism, nephrotic syndrome, hepatic impairment, acute pancreatitis) should also be ruled out.

## 9. FH Physical Examination

Patients with FH do not experience symptoms of the disease. The disease is diagnosed accidentally or in the presence of ACS [17]. When examining the patient, we can pay attention to the accumulation of cholesterol in the skin of the hands, toes, feet, elbows, and knees—looking for xanthomas. If a patient has xanthomas observed at a young age, we can suspect the case of a homozygous FH [40]. Moreover, during the physical examination, the physician should also examine for the possible thickening of the Achilles tendon, which is often found in patients with FH [41]. Thickening of the Achilles tendon also can be detected by ultrasound of the mentioned area [41].

Xanthomas and macrophage-induced inflammation in the joints could cause pain in joint projection and polyarthritis. Therefore, the anamnestic data of the painful joints could also help detect FH [42].

Moreover, examining the patient with FH, we also can find typical changes in the eyes. The deposition of lipids in the cornea can cause single corneal circle, called arcus cornealis [42]. FH and the excess of the cholesterol in the blood could cause cholesterol deposits in the eyelids (xanthelasmas).

During the cardiac ultrasound, calcified aortic valve is detected statistically more frequently in patients with FH [43]. This pathology is observed in almost 100% of patients with homozygous FH. Patients with heterozygous FH mostly have subclinical form or aortic valve stenosis [43]. Patients with FH predominantly does not have arterial hypertension, type 2 diabetes mellitus (DM), metabolic syndrome, or hyperuricemia (the incidence of these diseases is statistically lower than in the population) [44].

## 10. Cost-Effectiveness of Different Diagnostic Tools for FH Screening

FH screening within primary care is considered cost-effective. There are several studies that try to identify which screening method is the most cost-effective. Clinical effectiveness was measured based on identified number of monogenic FH cases. While screening for FH within primary care has been demonstrated to be cost-effective [45], which screening approach is cost-effective has not been established [46]. The model found that the addition of primary care case identification by database search for patients with recorded total cholesterol > 9.3 mmol/L was more cost effective than cascade testing alone [44]. A cost-effectiveness study in Poland showed that screening patients with acute coronary syndrome (ACS) younger than 55–65 years is the most cost-effective strategy. Removing age limit or using genetic testing has reduced cost-effectiveness [47]. A study in Spain showed that sequential implementation of genetic screening in patients relatives who was previously diagnosed with FH is also cost-effective [48]. There is no single approved FH screening program in our country, but based on the good experience of other countries, it is recommended to carry out screening according to the criteria of the Dutch Lipid Clinic Network or genetic screening program for the close relatives of individuals with FH.

## 11. Treatment

### 11.1. Non-Pharmacological Treatment of FH

FH treatment is multifaceted to reduce TC and LDL-C levels, and lifestyle correction is important [49]. The American Heart Association (AHA) promotes healthy eating skills to reduce the risk of IHD [50]. It is recommended to reduce the products of animal origin, and increase the quantities of vegetables, fruits, fiber, and cereals [51]. Portfolio and Mediterranean diets are best suited for the correction of hypercholesterolemia [50].

Physical activity (PA) is an important factor in controlling cholesterol levels because it helps the patient to reduce their body mass index and to reduce the incidence of acute cardiac ischemia [52]. A positive effect is found when PA is performed for at least 150–300 min per week of moderate intensity, or 75–150 min per week of vigorous intensity aerobic PA. [53]. Studies have shown that combining physical activity with a healthy diet and the use of dietary supplements (fish oil, plant sterols, oat bran) can reduce LDL-C levels by up to 30% [54].

Patients should also be encouraged to quit smoking, as smoking increases the risk of IHD and cardiac death [55].

### 11.2. Drug Treatment of FH

The main goal of FH treatment is to reduce plasma LDL-C levels [48]. In primary prevention for individuals at very high risk, an LDL-C reduction of more than 50% from baseline and an LDL-C goal of less than 1,4 mmol/L should be considered [56]. Seven groups of lipid-modifying drugs are currently available: statins, resins, fibrates, nicotinic acid derivatives, cholesterol reabsorption inhibitors, and monoclonal antibodies (PCSK9 inhibitors) and small-interfering-RNA-based therapeutic (inclisiran).

### 11.3. Statins

Statins are the drugs of first choice that can be administered at the highest tolerated doses [57]. Statins are hydroxymethylglutaryl (HMG) CoA reductase inhibitors, causing cholesterol synthesis inhibition, particularly intrahepatic cholesterol. This reduces LDL-C by ≥50% and TG levels by 10–20% from baseline values. [58]. Reduction in LDL-C is the main effect of statins, but statins also increase blood HDL-C levels (from 5 to 10%) [46]. Statins reduce the incidence of IHD, and are also useful for primary and secondary prevention in patients with high cardiovascular risk. High doses of statins promote the regression of atherosclerosis [57]. They also reduce vascular remodeling, inhibit the inflammatory response within the blood vessel, dilate the inner and middle layers of the artery, affect blood clotting, and contribute to the stabilization of the atherosclerotic plaque [59]. Statin treatment reduces the risk of dangerous complications of CVD: MI by 23%, CAD death by 20%, any stroke or coronary revascularization by 17%, and total mortality by 10%, over 5 years [60]. A ROMA II clinical trial, published in 2013, showed that patients pretreated with higher doses of statins before undergoing PCI had a lower risk of MACCE up to 12 months [61]. The extended ESTABLISH trial examined whether the early initiation of statin in patients with acute coronary syndrome (ACS) improves long-term prognosis. The results showed that early intensive statin therapy after ACS improved long-term prognosis (cumulative MACCE-free survival was significantly higher in the atorvastatin group than in the control group) [62]. Moreover, in randomized controlled trial, high-dose versus low-dose pitavastatin reduced MACE risk by 46% in patients in the highest baseline small dense LDL cholesterol [63].

### 11.4. Combination Therapy

The combination of a statin and a cholesterol absorption inhibitor is considered to be the first-line combination cholesterol-lowering therapy. Ezetimibe is currently the only drug in the class of cholesterol absorption inhibitors [64]. It selectively blocks the intestinal absorption of cholesterol and some phytosterols [65]. Ezetimibe reduces LDL-C by approximately 18% and slightly increases HDL-C [66].

The second combination option is a statin and bile acid sequestrant [67]. Bile acid sequestrates bind to bile acids and prevent them from entering the enterohepatic circulation, causing the liver to produce more bile acids to replace those lost. Because the body uses cholesterol to produce bile acids, this mechanism lowers LDL-C in the blood [53]. The use of bile acid (colesevelam, colestipol, or cholestyramine) sequestrants can reduce blood LDL-C levels by 15–30% [68].

The results of the IMPROVE-IT trial revealed that ezetimibe added to statins could further reduce LDL-C levels in blood. These findings of further LDL-C reduction mean that, starting with a baseline LDL-C of ≈70 mg/dL, a further 20% reduction in LDL-C translates into a 6% to 7% lower risk of MACE [69].

Moreover, the data analysis of the ODYSSEY trial revealed that in patients who consume maximally tolerated doses of statins and either alirocumab or ezetimibe, for every additional 39 mg/dL lower LDL-C level achieved, there was a further 24% lower risk of MACE [70].

### 11.5. PCSK9 Inhibitors

PCSK9 is an enzyme produced by hepatocytes that binds to low-density lipoprotein receptors (LDL-R) on the surfaces of hepatocytes and promotes the breakdown of these receptors. PCSK9 inhibitors bind to the PCSK9 enzyme and promote its degradation, thereby increasing the amount of LDL-R on the surface of hepatocytes and ensuring more intensive removal of LDL-C from the blood plasma [71]. PCSK9 inhibitors reduce LDL-C levels by 67% and lipoprotein (a) (Lp (a)) by 20–36%. PCSK9 inhibitors are used when high LDL-C levels are maintained during treatment with maximal statin doses and there is a high risk of cardiovascular complications [72]. Their use should be considered in patients with FH and an intolerance of statins or high Lp (a) concentrations [71]. In the homozygous form of FH, PCSK9 inhibitors help to reduce LDL-C levels only if low-density lipoprotein receptor (LDLR) expression is maintained [73]. A recent clinical trial that reviewed ODYSSEY OUTCOMES showed that the PCSK9 inhibitor alirocumab reduced the risk of MACE compared with a placebo, with an HR of 0.85 [74]. A randomized control study that examined the effects of PCSK-9 inhibitor evolocumab, additional to statin treatment, in high-risk patients with stable atherosclerosis showed that this combination reduced the risk of ischemic stroke by 0.4% with no increase in hemorrhage stroke [75].

### 11.6. Nicotinic Acid (Vitamin B3)

Nicotinic acid inhibits lipolysis in adipose tissue, resulting in less free fatty acids entering the liver. This reduces the synthesis of TG required for the production of LDL-C in the liver. As very-low-density lipoprotein levels decrease, blood LDL-C levels also decrease [76]. They are rarely used, especially in children, due to the risk of poor tolerance, myopathy, hyperuricemia, and hepatitis [73].

### 11.7. Fibrates

Fibrates are derivatives of fibric acid that are used to treat hypertriglyceridemia. When TG levels are normal, fenofibrate reduces LDL-C by 15–20%, and can increase HDL-C by up to 20% [76].

### 11.8. Inclisiran

Inclisiran is a first-in-class, cholesterol-lowering small interfering RNA (siRNA) conjugated to triantennary N-acetylgalactosamine carbohydrates (GalNAc) [77]. It is used to treat heterozygous familial and non-familial or mixed dyslipidemia. Inclisiran reduces LDL-C levels by inhibiting the hepatic translation proprotein convertase subtilisin/kexin type 9 (PCSK9), thereby upregulating the number of LDL-receptors on the hepatocytes [77]. Inclisiran reduces LDL-C by over 50% with one dose every 6 months [60].

### 11.9. Treatment of Homozygous FH with Mipomersen and Lomitapide

In adults with severe homozygous FH, mipomersen and lomitapide are used as an adjunctive therapy in combination with standard FH therapy [78]. Mipomersen is an oligonucleotide that binds to ApoB100 mRNA and arrests the translation of this protein, thereby reducing the synthesis of ApoB100-containing lipoproteins and the production of VLDL and LDL. Mipomersen causes a dose-dependent decrease in plasma levels of LDL-C, ApoB 100, Lp (a), and TG [79].

Lomitapide is an inhibitor of microsomal triglyceride transport protein (MTP). MTP is involved in the synthesis and secretion of ApoB-containing lipoproteins in the liver and intestine; therefore, by reducing the synthesis of these lipoproteins, the concentration of LDL-C decreases. Lomitapide in homozygous FH can reduce LDL-C levels by 40% of their initial concentration [80].

### 11.10. Lipoprotein Apheresis

Lipoprotein apheresis should be used in patients at high risk of cardiovascular events and when medication does not achieve the prescribed LDL-C targets [81]. LDL apheresis reduces the amount of lipoprotein-containing ApoB in the blood [82]. This treatment (every 1–2 weeks) can decrease plasma LDL-C levels by 50–75%. [82]. Patients with higher baseline cholesterol levels have been found to have an even greater response to this treatment.

### 11.11. Other Treatment Possibilities

In recent years, attempts have been made to apply gene therapy in FH treatment. In the preclinical research phase, gene therapy, stem cell transplantation, and autologous transplantation of genetically “corrected” cells are evaluated [83].

Cholesteryl ester transfer protein (CETP) inhibitors are among the treatment options for the future. CETP inhibitors inhibit cholesteryl ester transfer protein, which normally transfers cholesterol from HDL cholesterol to VLDL or LDL-C [84]. Inhibition of this process results in higher HDL-C levels and reduces LDL-C levels. [84]. They reduce LDL-C levels by 30–40% [67].

## 12. Summary

FH is a severe genetic disorder that often affects very young people. Early diagnosis and optimal treatment of this disease are extremely important factors for preventing further development of IHD. Genetic testing helps to differentiate patients with heterozygous or homozygous FH, as well as to identify family members with this pathology through stepwise family screening. However, phenotypical evaluation and lipid diagram are also crucial in completing a patient’s risk profile. Treatment of FH is complex, consisting of non-pharmacological therapy, drug therapy, and lipid apheresis, if conservative therapies do not help.

## Figures and Tables

**Table 1 medicina-58-01665-t001:** The Dutch Lipid Clinic Diagnostic Criteria for FH detection. Adapted with permission from Ref. [33]. 2004 Sept, Melissa A. Austin et al.

	Criteria	Score
Family history	First degree relative with premature CAD and/or LDL-C > 95‰	1
	First degree relative with tendon xanthomas and/or children under 18 years old with LDL-C > 95‰	2
Clinical history	Premature CAD	2
	Premature cerebral/peripheral vascular disease	1
Physical examination	Tendon xanthomas	6
	Arcus cornealis < 45 years old	4
LDL-C	>8.5 mmol/L	8
	6.5–8.4 mmol/L	5
	5.0–6.4 mmol/L	3
	4.0–4.9 mmol/L	1
Definite FH		>8
Probable FH		6–8
Possible FH		3–5
Unlikely FH		<3

## Data Availability

The data are available upon request from the author.

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
