# Peer review of "Familial Hypercholesterolemia and Its Current Diagnostics and Treatment Possibilities: A Literature Analysis"

_medicina, 2022, doi:10.3390/medicina58111665_

Round 1
Reviewer 1 Report
I appreciate the opportunity given to review this manuscript.Familial hypercholesterolemia (FH), an autosomal dominant genetic disease that is increasingly emerging as a global threat and is currently of great interest.
The future direction for his research is on what are the mechanisms of FH and its complications, its diagnosis, its statin therapy and on new lipid-lowering drug therapy. In this manuscript, a very good updated review of the different topics of interest was carried out.
My recommendations are the following to strengthen your content regarding the section on its epidemiology:
1. I think that the epidemiology section can be explored a little more to give a better global vision, hopefully by continent, of its prevalence as well as its participation in ischemic heart disease.
Here some references
1.Candace L. Jackson, Magdi Zordok, Iftikhar J. Kullo,Familial hypercholesterolemia in Southeast and East Asia,American Journal of Preventive Cardiology,Volume 6,2021,100157, ISSN 2666-6677, https://doi.org/10.1016/j.ajpc.2021.100157.
2. Meshkov AN, Ershova AI, Kiseleva AV, Shalnova SA, Drapkina OM, Boytsov SA, on behalf of the FH-ESSE-RF Investigators. The Prevalence of Heterozygous Familial Hypercholesterolemia in Selected Regions of the Russian Federation: The FH-ESSE-RF Study. Journal of Personalized Medicine. 2021; 11(6):464. https://doi.org/10.3390/jpm11060464
3. Familial hypercholesterolemia in Mexico: Initial insights from the national registry, Journal of Clinical Lipidology,Volume 15, Issue 1,2021,Pages 124-133,ISSN 1933-2874,https://doi.org/10.1016/j.jacl.2020.12.001.
4.Seyedmohammad Saadatagah, Lubna Alhalabi, Medhat Farwati, Magdi Zordok, Ashwini Bhat, Carin Y. Smith, Christina M. Wood-Wentz, Kent R. Bailey, Iftikhar J. Kullo,The Burden of Severe Hypercholesterolemia and Familial Hypercholesterolemia in a Population-based Setting in the US,American Journal of Preventive Cardiology,2022,100393,ISSN 2666-6677, https://doi.org/10.1016/j.ajpc.2022.100393.
Reviewer 2 Report
This manuscript is an excellent summary of FH, however I feel several lack of the required statement.
In hypercholesterolemia, most importance is, “How do FH associate to increase the risk of major cerebro-cardiovascular event (MACCE) such as ischemic heart disease or stroke and how we should treat the risk”.
Hence the following points should be added;
How much do FH and serum LDL level increase the risk of MACCE?
How much do current medical treatment, especially in statins, PCSK9, and ezetimibe, decrease the risk of MACCE.
Representative large scale clinical studies regarding clinical efficacy of the above medications.
Reviewer 3 Report
Dear Editor,
Familial hypercholesterolaemia has great potential in the prevention of premature disease and death from CAD. Despite advances in knowledge, there is still a significant shortcoming in the detection and treatment of this relatively common hereditary disease. This review provides an overview of FH but does not present the latest advances in the care of FH for the prevention of CAD in affected families. Key advances in FH include genetics, imaging, registries, and therapy.
In my opinion, this work is a recommendation for physicians in the diagnosis and treatment of patients with FH rather than a review and analysis of current literature. I suggest that the manuscript be published in a national Lithuanian medical journal as a guide for doctors.
In addition, the literature prepared by the authors does not meet the criteria adopted for this journal. The literature should be fully verified in accordance with the guidelines in the guide for authors.
Reviewer 4 Report
1. use rather evolocumab and alirocumab than evocumab and alicumab (line 19)
2. change "reduction of ... death" into "reduction of ... mortality" (line 21)
3. acute coronary disease? do you mean acute coronary syndrome? (line 23, 119)
4. first lipidogram? - naive lipidogram
5. explain the abbreviation AH - line 136
6. with FH intolerance - should be " with FH and intolerance" - line 197
7. I suggest adding a paragraph about the cost-effectiveness of different diagnostic tools for FH screening
8. I suggest adding a table with the most frequent genetic variants detected in patients with FH as well as information about geographical/regional differences in variants of LDLR, APOB, and PCSK9 genes
9. check grammatical correctness (lines 66, 136, 189)
Round 2
Reviewer 3 Report
I accept the changes and additions made by the Authors in the manuscript "Familial hypercholesterolaemia and its current diagnostics and treatment possibilities: Literature analysis". The current version contains elements of novelty, especially in the aspect of treating familial hypercholesterolemia.
I suggest accepting the article for publication.
Reviewer 4 Report
after some little correction the manuscript is ready for publication:
line 22 - missing gap between words
line 66 - I suggest "concomitant" instead of "additional"
line 189 - "were" instead of "was"